# Host phenotype classification from human microbiome data is mainly driven by the presence of microbial taxa

**Renato Giliberti[1], Sara Cavaliere[1], Italia Elisa Mauriello[1], Danilo Ercolini[1,2], Edoardo Pasolli**[1,2]*

**1** Department of Agricultural Sciences, University of Naples Federico II, Portici, Italy, **2** Task Force on Microbiome Studies, University of Naples Federico II, Naples, Italy

* edoardo.pasolli@unina.it

## Abstract

Machine learning-based classification approaches are widely used to predict host phenotypes from microbiome data. Classifiers are typically employed by considering operational taxonomic units or relative abundance profiles as input features. Such types of data are intrinsically sparse, which opens the opportunity to make predictions from the presence/absence rather than the relative abundance of microbial taxa. This also poses the question whether it is the presence rather than the abundance of particular taxa to be relevant for discrimination purposes, an aspect that has been so far overlooked in the literature. In this paper, we aim at filling this gap by performing a meta-analysis on 4,128 publicly available metagenomes associated with multiple case-control studies. At species-level taxonomic resolution, we show that it is the presence rather than the relative abundance of specific microbial taxa to be important when building classification models. Such findings are robust to the choice of the classifier and confirmed by statistical tests applied to identifying differentially abundant/present taxa. Results are further confirmed at coarser taxonomic resolutions and validated on 4,026 additional 16S rRNA samples coming from 30 public case-control studies.

**Data Availability Statement:** The data and source code used to produce the results and analyses presented in this manuscript are available on a

## Author summary

The composition of the human microbiome has been linked to a large number of different diseases. In this context, classification methodologies based on machine learning approaches have represented a promising tool for diagnostic purposes from metagenomics data. The link between microbial population composition and host phenotypes has been usually performed by considering taxonomic profiles represented by relative abundances of microbial species. In this study, we show that it is more the presence rather than the relative abundance of microbial taxa to be relevant to maximize classification accuracy. This is accomplished by conducting a meta-analysis on more than 4,000 shotgun metagenomes coming from 25 case-control studies and in which original relative abundance data are degraded to presence/absence profiles. Findings are also extended to

GitHub repository at https://github.com/RGilib/giliberti-meta-analysis-2022.

**Funding:** The work was by supported by P.O.R. Campania FSE 2014/2020 to R.G. The funder had no role in study design, data collection and analysis, decision to publish, or preparation of the manuscript.

**Competing interests:** The authors have declared that no competing interests exist.

16S rRNA data and advance the research field in building prediction models directly from human microbiome data.

This is a *PLOS Computational Biology* Methods paper.

## Introduction

Evidence has linked the human microbiome, the large set of microorganisms that reside in our body, with health and disease conditions [1]. Several diseases have been associated with microbiome traits and estimation of host phenotypes from microbiome composition has received remarkable attention in the community. In this regard, growing attention has been given to predicting host phenotypes using machine-learning based approaches, and in which adoption of classification methodologies for case-control studies has represented the most investigated scenario [2]. Classification represents a practical approach to implicitly integrate multiple characteristics (i.e., features; such as the case of combination of hundreds of microbial relative abundances) and get evaluation metrics of relatively easy interpretation. This is the case of the area under the receiver operating characteristic curve (AUC), the most used metric in the microbiome field for binary classification problems [2], which ranges in value from 0 to 1 with better accuracy when moving towards one.

Focusing on case-control studies, machine learning methods have been involved in two main types of analyses. The first has relied on applying established methodologies to newly generated data, which has allowed researchers to provide evidence of the predictability of host phenotypes from microbiome data for several different diseases including inflammatory bowel disease [3], obesity [4], type-2 diabetes [5], colorectal cancer [6], and paved the way to the potential use of the microbiome as a diagnostic tool [7,8]. The increasing number of large population studies [9,10] has also enabled the implementation of several (large-scale) meta-analyses aiming at validating findings across independent cohorts. Besides analyses based on 16S rRNA data [11–13], similar efforts have been extended more recently to shotgun data [14–17], while extension to other-omics data has been more challenging [18]. The second group of analyses has been focused on the proposal of new methodologies in two main directions: extraction of better feature representations or optimization at classifier level [19]. While classification can be applied on the original set of features, improvements can be obtained by reducing the dimensionality of the feature space (for example by selecting or extracting specific operational taxonomic units (OTUs) or microbial taxa). Examples include feature subset selection [20], recursive feature elimination [14], and hierarchical feature engineering [21]. Different (supervised) methods have been adopted for classification purposes. Some widely used strategies are represented by logistic regression [22], support vector machines (SVMs) [3], k-nearest neighbours [23], and random forests (RFs) [14]. Comparisons among different classifiers have also been performed, with ensemble methods such as RFs and extreme gradient boosting decision trees that have exhibited in general the best performances [24]. Recently, different solutions based on deep learning approaches have been also proposed [25,26], including methods to transform high-dimensional data into robust low-dimensional representations [27], although challenges still arise due to the limited amount of labelled information that is typically available in case-control microbiome studies [28].

Despite the different methodologies adopted along the classification pipeline, classification models have been typically built by considering OTU or relative abundance profiles as input features. However, such types of data are intrinsically sparse, therefore this potentially enables to make inferences from the presence/absence of microbial taxa rather than their relative abundance values. This also poses the question whether it is the presence of particular taxa rather their abundance values to be relevant for discrimination purposes. Surprisingly, this aspect has not been investigated yet. In this paper, we aim at filling this gap by presenting a meta-analysis on publicly available datasets from both shotgun and 16S rRNA data.

## Materials and methods

### The considered publicly available metagenomic and 16S rRNA datasets

In this paper, we conducted a meta-analysis on publicly available human metagenomic datasets for host phenotype classification. More specifically, we considered 4,128 samples coming from 25 shotgun metagenomic studies/datasets as summarized in **Table 1** and **Fig 1A**. Twenty-one studies were devoted to the characterization of the gut microbiome in association with different diseases (i.e., case-control studies). Two additional datasets were case-control studies (peri-implantitis, mucositis, and schizophrenia) from oral metagenomes. We also considered a dataset aiming at characterizing changes in the human microbiome due to

**Table 1. Summary of the 25 classification tasks derived from metagenomic datasets for case-control prediction.** ACDV: Atherosclerotic cardiovascular disease, AD: Alzheimer's disease, BD: Behcet's disease, CRC: Colorectal cancer, IBD: irritable bowel disease, T1D: Type 1 diabetes, T2D: Type 2 diabetes. We additionally considered the HMP_2012 dataset [10] for body site discrimination between gut (N = 414) and oral (N = 147) samples.

| Dataset name | Body site | # controls | Cases | # cases | Reference |
|---|---|---|---|---|---|
| JieZ_2017 | Gut | 171 | ACVD | 214 | [31] |
| ChngKR_2016 | Skin | 40 | AD | 38 | [32] |
| YeZ_2018 | Gut | 45 | BD | 20 | [33] |
| RaymondF_2016 | Gut | 36 | Cephalosporins | 36 | [34] |
| QinN_2014 | Gut | 114 | Cirrhosis | 123 | [35] |
| FengQ_2015 | Gut | 61 | CRC | 46 | [36] |
| GuptaA_2019 | Gut | 30 | CRC | 28 | [37] |
| HanniganGD_2017 | Gut | 28 | CRC | 27 | [38] |
| ThomasAM_2018a | Gut | 24 | CRC | 29 | [39] |
| ThomasAM_2018b | Gut | 28 | CRC | 32 | [39] |
| VogtmannE_2016 | Gut | 52 | CRC | 52 | [40] |
| WirbelJ_2018 | Gut | 65 | CRC | 60 | [41] |
| YachidaS_2019 | Gut | 251 | CRC | 258 | [42] |
| YuJ_2015 | Gut | 53 | CRC | 75 | [43] |
| ZellerG_2014 | Gut | 54 | CRC | 61 | [6] |
| LiJ_2017 | Gut | 41 | Hypertension | 99 | [44] |
| IjazUZ_2017 | Gut | 38 | IBD | 56 | [45] |
| NielsenHB_2014 | Gut | 248 | IBD | 148 | [46] |
| GhensiP_2019_m | Oral | 49 | Mucositis | 20 | [47] |
| GhensiP_2019 | Oral | 49 | Peri-implantitis | 23 | [47] |
| Castro_NallarE_2015 | Oral | 16 | Schizophrenia | 16 | [48] |
| Heitz-BuschartA_2016 | Gut | 26 | T1D | 27 | [49] |
| KosticAD_2015 | Gut | 89 | T1D | 31 | [50] |
| KarlssonFH_2013 | Gut | 43 | T2D | 53 | [51] |
| QinJ_2012 | Gut | 174 | T2D | 170 | [52] |

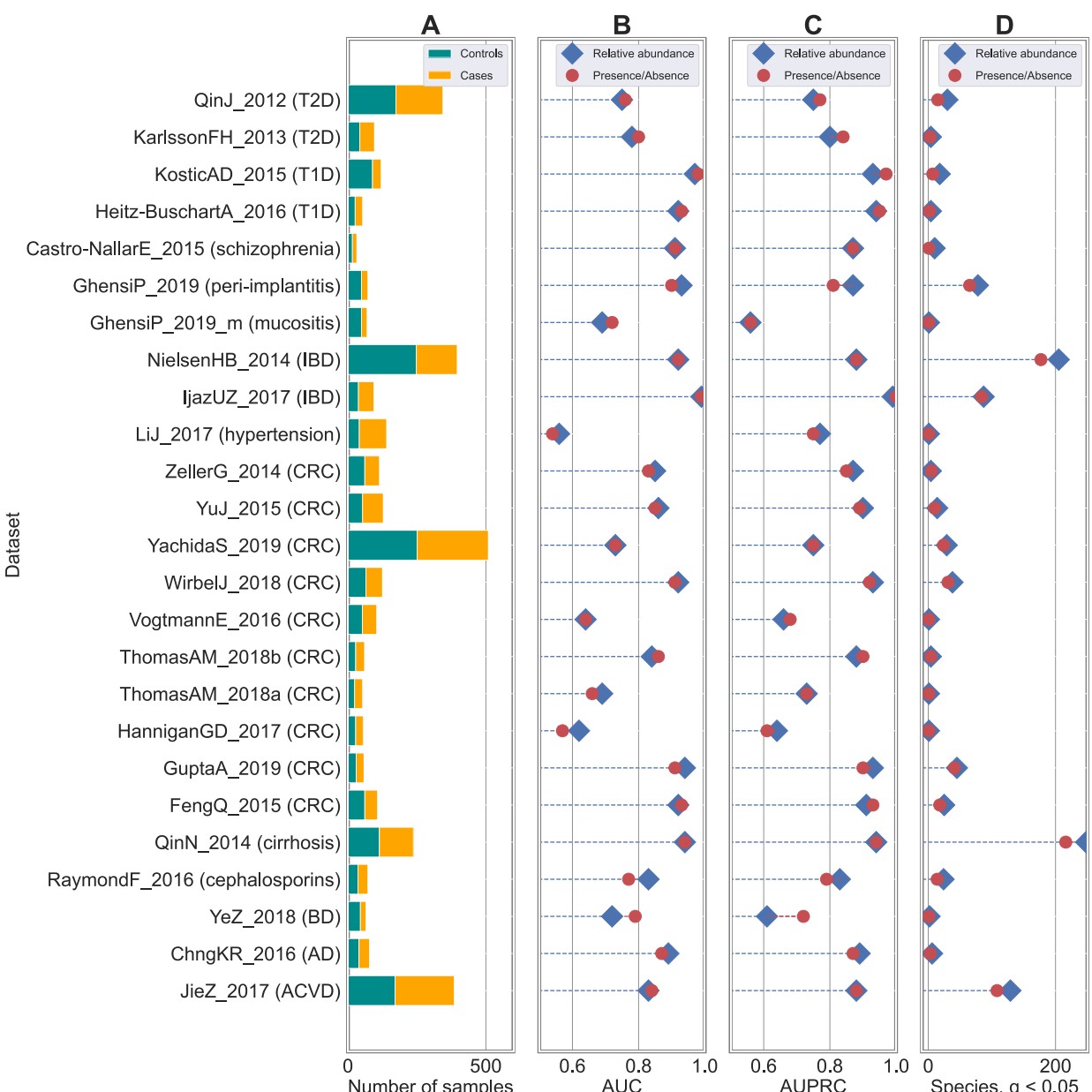

**Fig 1. Classification accuracies are robust to degradation from species-level relative abundance to presence/absence profiles in shotgun datasets.** Results obtained on 25 case-control studies for host phenotype classification from human microbiomes. (**A**) Number of case and control samples across the different studies. (**B**) AUC and (**C**) AUPRC scores using RF as back-end classifiers on species-level taxonomic profiles. Comparison between relative abundance (in blue) and presence/absence (in red) profiles highlighted negligible differences and no statistical differences in none of the studies (see **S1 Fig** for AUC scores and **S2 Table** for p-values). Metrics of comparison in terms of AUC, AUPRC, precision, recall, and F1 are summarized in **S2 Fig and S2 Table** is represented a comparison between **AUC** and **AUPCR** scores. (**D**) Number of statistically significant taxa from relative abundance (in blue) and presence/absence (in red) profiles.

consumption of cephalosporins, while the last dataset was devoted to the discrimination between body sites (i.e., stool vs oral) in the Human Microbiome Project (HMP) dataset. Metagenomic samples were processed to generate species-level taxonomic profiles through MetaPhlAn3 [29]. Species abundances are expressed as real numbers in the range [0,1] with values that sum to 1 for each sample. Generation of relative abundances at other taxonomic

levels (i.e., genus, family, and order) was also extracted from the MetaPhlAn3 output. Metadata information in terms of disease status or body site for the HMP dataset are available in the curatedMetagenomicData package [30].

We additionally analysed 4,026 16S rRNA samples coming from 30 publicly available case-control studies (**S1 Table** and **Fig 2A**). We considered the same set of gut samples considered

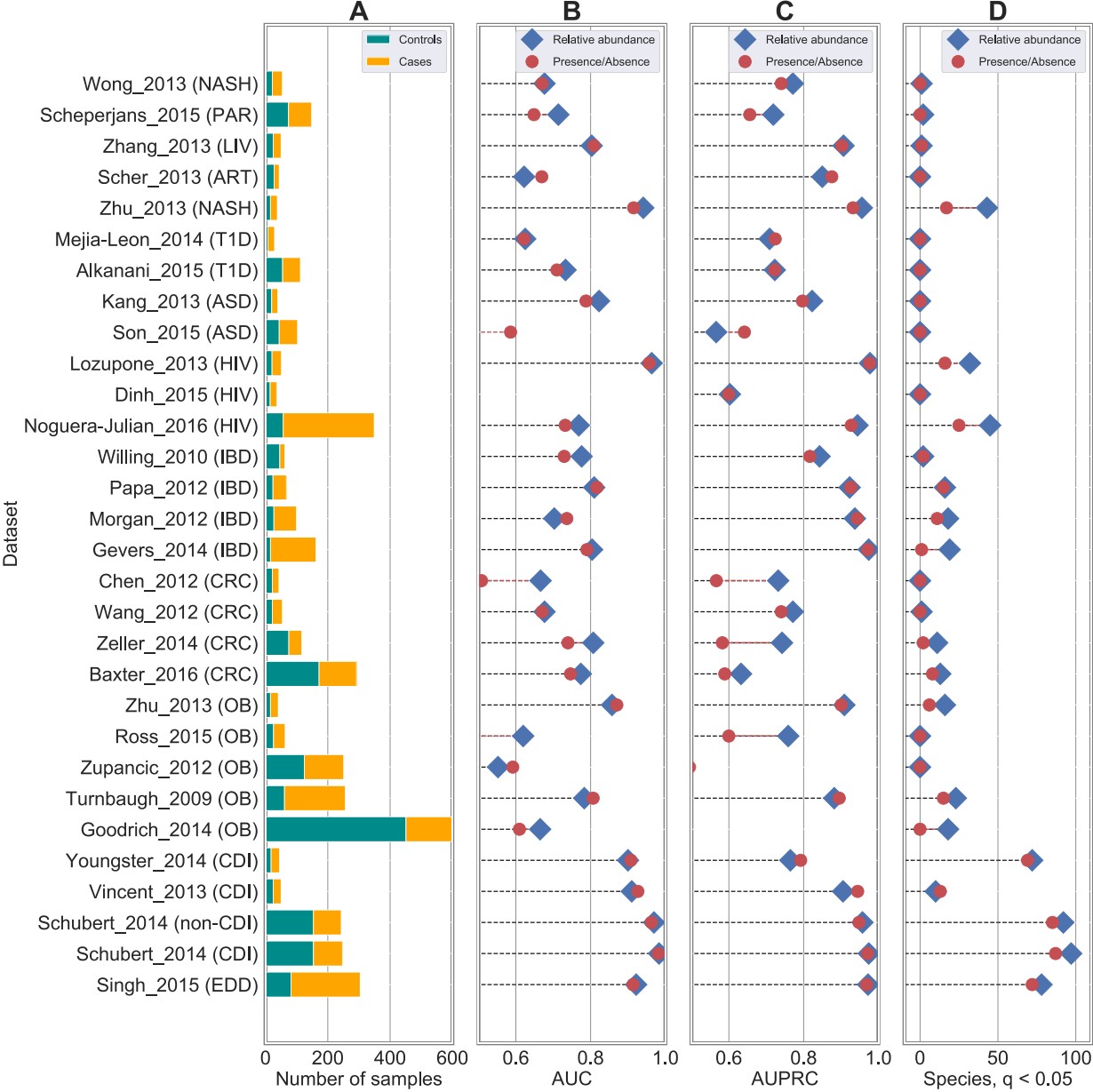

**Fig 2. Classification accuracies are robust to degradation from genus-level relative abundance to presence/absence profiles in 16S rRNA datasets.** Results obtained on 30 case-control studies for host phenotype classification from human microbiomes. (**A**) Number of case and control samples across the different studies. (**B**) AUC and (**C**) AUPRC scores using RF as back-end classifiers on species-level taxonomic profiles. Comparison between relative abundance (in blue) and presence/absence (in red) profiles highlighted negligible differences and no statistical differences in none of the studies (see **S5 Table** for p-values) as found also in shotgun datasets (see **Fig 1**). Metrics of comparison in terms of AUC, AUPRC, precision, recall, and F1 are summarized in **S5 Table**. (**C**) Number of statistically significant taxa from relative abundance (in blue) and presence/absence (in red) profiles.

in [13] with metadata information in terms of disease status as follows: autism spectrum disorder (ASD), *Clostridioides difficile* infection (CDI), CRC, enteric diarrheal disease (EDD), human immunodeficiency virus (HIV), IBD, liver cirrhosis (CIRR), minimal hepatic encephalopathy (MHE), non-alcoholic steatohepatitis (NASH), obesity (OB), Parkinson disease, psoriatic arthritis (PSA), rheumatoid arthritis (RA), and T1D. 16S rRNA samples were preprocessed following the same procedure adopted in [13]. More specifically, we discarded samples with fewer than 100 reads and removed OTUs with less than 10 reads and/or present in less than 1% of the samples. After calculating the relative abundance of each OTU, OTUs were collapsed to genus level by summing their relative abundance values and by discarding any OTUs which were un-annotated at the genus level.

## The adopted machine learning methods

The classification tasks on both shotgun and 16S rRNA data were carried out by considering the already developed and validated MetAML (*Met*agenomic prediction *An*alysis based on *M*achine *L*earning) tool [14]. Main analyses were conducted by using Random Forests (RFs) as back-end classifiers, and validations were extended to other three classifier types: support vector machines with linear (denoted with LSVM in this paper) and RBF (denoted with SVM in this paper) kernel, Lasso, and Elastic Net (ENet).

Free parameters of the classifiers were set as follows. For RF, i) the number of trees was set to 500, ii) the number of features to consider when looking for the best split was equal to the root of the number of original features, and iii) the gini impurity criterion was used to measure the quality of a split. For Lasso and ENet, the regularization parameters were obtained using a 5-fold stratified cross-validation approach. For Lasso the alpha parameter was found in the set $\{10^{-4}, \ldots, 10^{-0.5}\}$ with 50 uniform steps. For ENet, besides the alpha parameter, also the L1_ratio parameter was chosen in the set [0.1, 0.5, 0.7, 0.9, 0.95, 0.99, 1.0].

## Validation and evaluation strategies

We conducted two main types of analysis: i) cross-validation and ii) cross-study analysis. In cross-validation, samples were randomly divided into $k$ (with k = 10 in our case) folds by considering a stratified cross-validation approach to preserve the percentage of samples of each class. Results were repeated and averaged on 20 independent runs. Different models were trained on the same cross-validation splits. We also considered a cross-study analysis in order to evaluate robustness of the prediction when transferring models from a source to a target domain. In this setting, the classification model was trained on the source dataset and accuracy was evaluated on a different independent dataset.

Classification accuracies were evaluated in terms of five main metrics: area under the curve (AUC), area under the precision-recall curve (AUPRC), precision, recall, and F1.

We calculated mean difference and standard error for each 10-fold CV and averaged across the 20 repetitions. We calculated the 95% confidence interval on the difference in AUC performance between two classifiers as done in [14] using the t-distribution with df = 9:

$$95\%CI: \quad \frac{1}{20}\frac{1}{10}\sum_{j=1}^{20}\sum_{i=1}^{10}(AUC_{1ij} - AUC_{2ij}) \pm 2.26 \, x \frac{\sigma_j}{\sqrt{10}} \tag{1}$$

where AUC1ij and AUC2ij are the AUC of two classifiers in fold i of repetition j, and σj is the standard deviation of the AUC1ij−AUC2ij across i = 1. . .10 folds in repetition j. We computed the p-values from the t-statistics from mean difference and standard error flatten over the 20

repetitions:

$$t = \frac{\frac{1}{20}\frac{1}{10}\sum_{j=1}^{20}\sum_{i=1}^{10}(AUC_{1ij} - AUC_{2ij})}{\frac{1}{20}\sum_{j=1}^{20}\frac{\sigma_j}{\sqrt{10}}}$$

(2)

We used a two-tailed t-test with df = 9.

## Experimental setting for shotgun datasets

Most of the analyses on shotgun datasets were conducted by considering a cross-validation approach. Twenty-four classification tasks were devoted to the discrimination of healthy from diseased subjects (i.e., case-control studies), while the HMP dataset was used to perform body site discrimination between gut and oral samples. We also considered the ten independent datasets associated with CRC and evaluated prediction capabilities in a cross-study setting.

Baseline results were obtained by considering the original relative abundance profiles at species-level resolution provided by MetaPhlAn3 [29] as features and using RF as back-end classifier. This is the setting that was successfully deployed and validated in multiple meta-analyses such as the ones presented in [14,30,39,47]. At this point, multiple comparisons were performed: i) starting from the original species-level relative abundance profiles (one profile for each sample), we generated presence/absence profiles by simply thresholding the relative abundance values at 0%. This generated a set of boolean profiles where 1 indicated the presence of the species regardless of its relative abundance in the considered sample, while 0 was associated with its absence. The same approach based on RF was applied on this set of newly generated profiles and compared with the results obtained on the original relative abundances. Results are summarized in **Figs 1B, 1C**, **S1 and S2**; ii) the same procedure described in i) was applied again by thresholding the relative abundance profiles at different values to assess sensitivity of classification to low abundant species. We considered these values as threshold levels: 0.0001%, 0.001%, 0.01%, and 0.1%. Results using RF as classifier are summarized in **Figs 3A and S3A**; iii) we extended the comparison done at species-level between original relative abundance and boolean (with threshold = 0%) profiles to three other taxonomic levels (i.e., genus, family, and order) to evaluate sensitivity of classification when moving from species to coarser taxonomic resolutions. Results with RF classification are summarized in **Figs 4 and S3B**; iv) we finally assessed robustness of our findings to the choice of the classification method. We compared RF results with the ones obtained by other four classifier algorithms (i.e., SVM with linear kernel, SVM with RBF kernel, Lasso, ENet) for both relative abundance and presence/absence profiles (**Figs 5 and S3C**). While we report in main figures only comparisons in terms of AUC, comparisons for the other three metrics (i.e., precision, recall, and F1) are reported in **S2 Table**.

## Experimental settings for 16S rRNA datasets

For 16S rRNA datasets we carried out only cross-validation analyses. From the genus-level profiles generated as described in the section "The considered publicly available metagenomic and 16S rRNA datasets", we generated the boolean profiles (with threshold = 0%) as similarly done for shotgun data. We compared the two types of profiles using a RF classifier (results in **Figs 2B, 2C** and **S4**), and were then extended also to the other classifier types (results in **S3 Table**).

## Statistical tests

On the same set of scenarios in which we compared classification accuracies, we conducted statistical tests to evaluate to which extent degradation from relative abundance to boolean profiles can impact the identification of differentially abundant species. We used Mann-Whitney U test

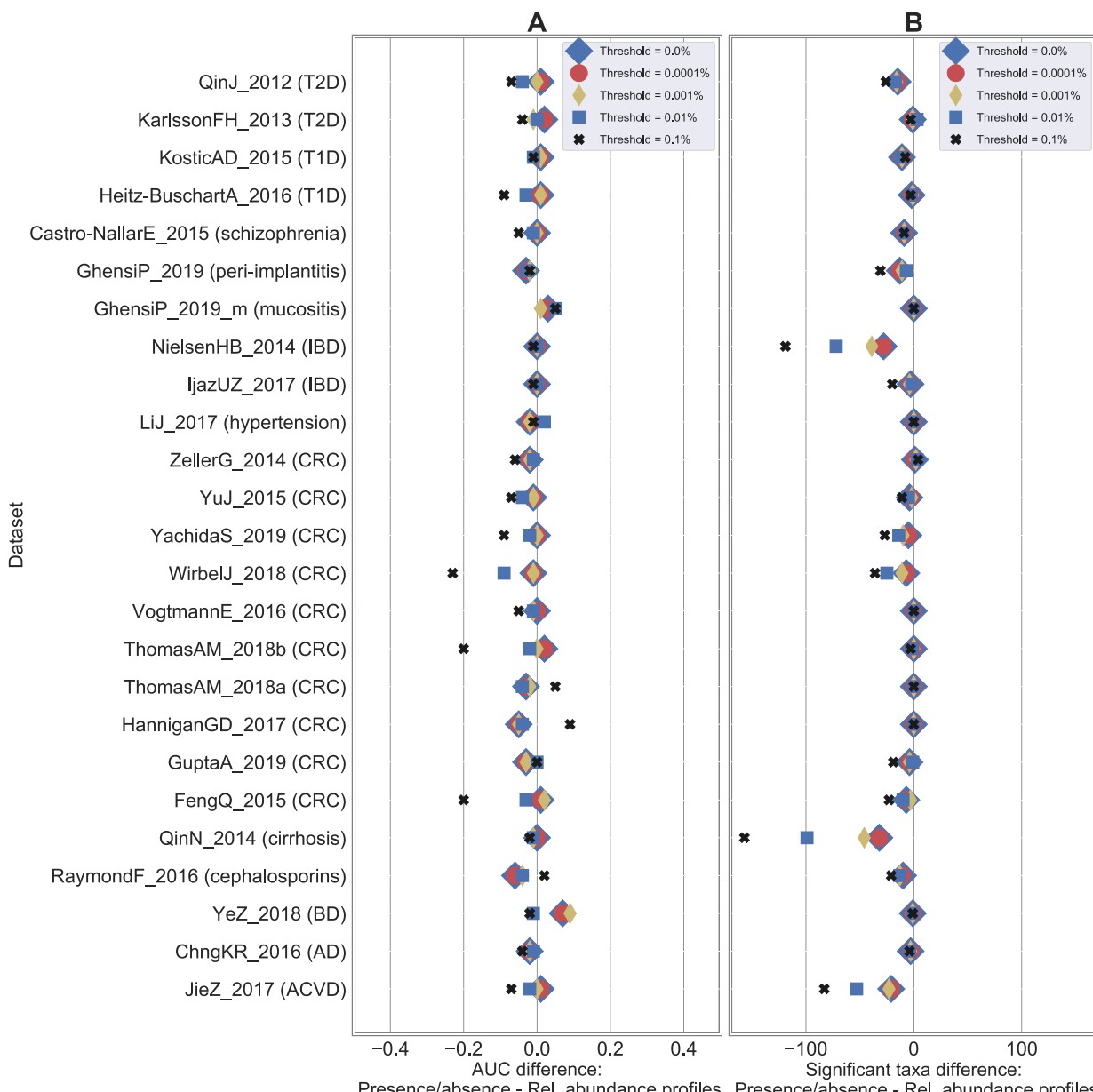

**Fig 3. Classification accuracies are not impacted when relative abundances are thresholded up to 0.001%.** Results on the 25 case-control shotgun studies by comparing the baseline (i.e., species-level relative abundance profiles) with the presence/absence profiles generated by thresholding at different relative abundance values (ranging from 0% to 0.1%). (A) Difference in AUC between the presence/absence and the relative abundance RF classification result. A positive value indicates that presence/absence outperforms relative abundance data. AUC scores at different thresholds are summarized in S2 Table. (B) Difference in number of statistically significant taxa (numbers summarized in S7 Table).

to identify the set of significant taxa when relative abundance profiles were involved, while we adopted Fisher exact test to deal with presence/absence data. Although it is out of the scope of the present study to perform a comprehensive evaluation of available statistical tests, further investigation taking into account alternatives including methodologies that can deal with compositional issues [53,54] is warranted. Finally, false detection rate (FDR) was applied for multiple testing correction, and corrected p-values < 0.05 identified significant taxa.

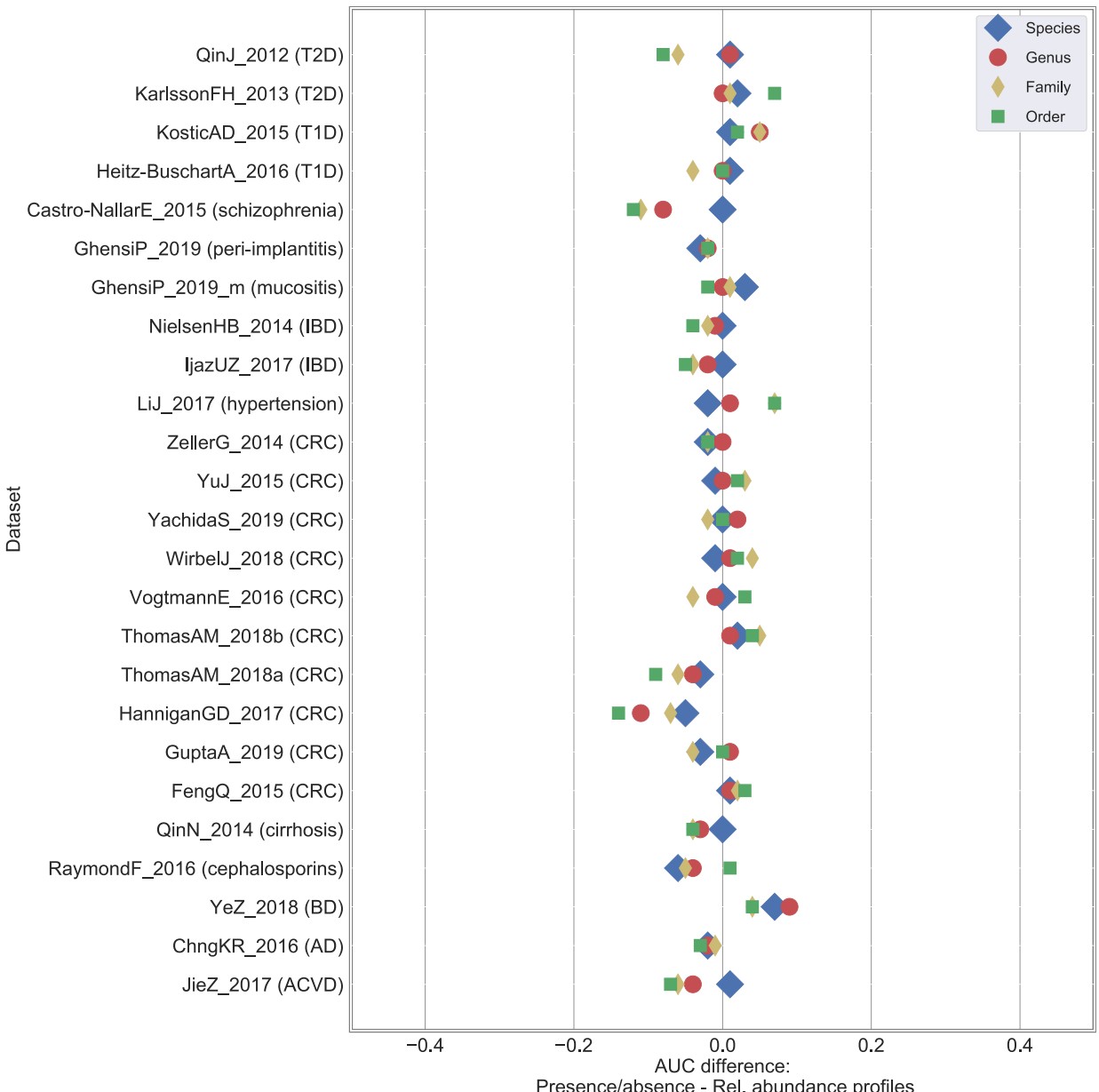

**Fig 4. Classification results are more impacted to relative abundance degradation at coarser taxonomic resolution.** Results on the 25 case-control shotgun studies by comparing the baseline (i.e., relative abundance profiles) with the presence/absence profile generated by thresholding at 0.0% and varying taxonomic resolution from species to order level. Difference in AUC between the presence/absence and the relative abundance RF classification result. A positive value indicates that presence/absence outperforms relative abundance data.

### Rarefaction analysis

We further performed rarefaction analysis by: i) considering the three datasets having the highest number of significant species from relative abundance profiles (i.e., JieZ_2017, NielsenHB_2014, and QinN_2014); ii) rarefying raw reads (using https://github.com/lh3/seqtk) and considering 1M reads for each metagenome; iii) applying the same pipeline to generate taxonomic profiles through MetaPhlAn3; iv) applying the same pipeline to build classification models and identifying statistically significant species.

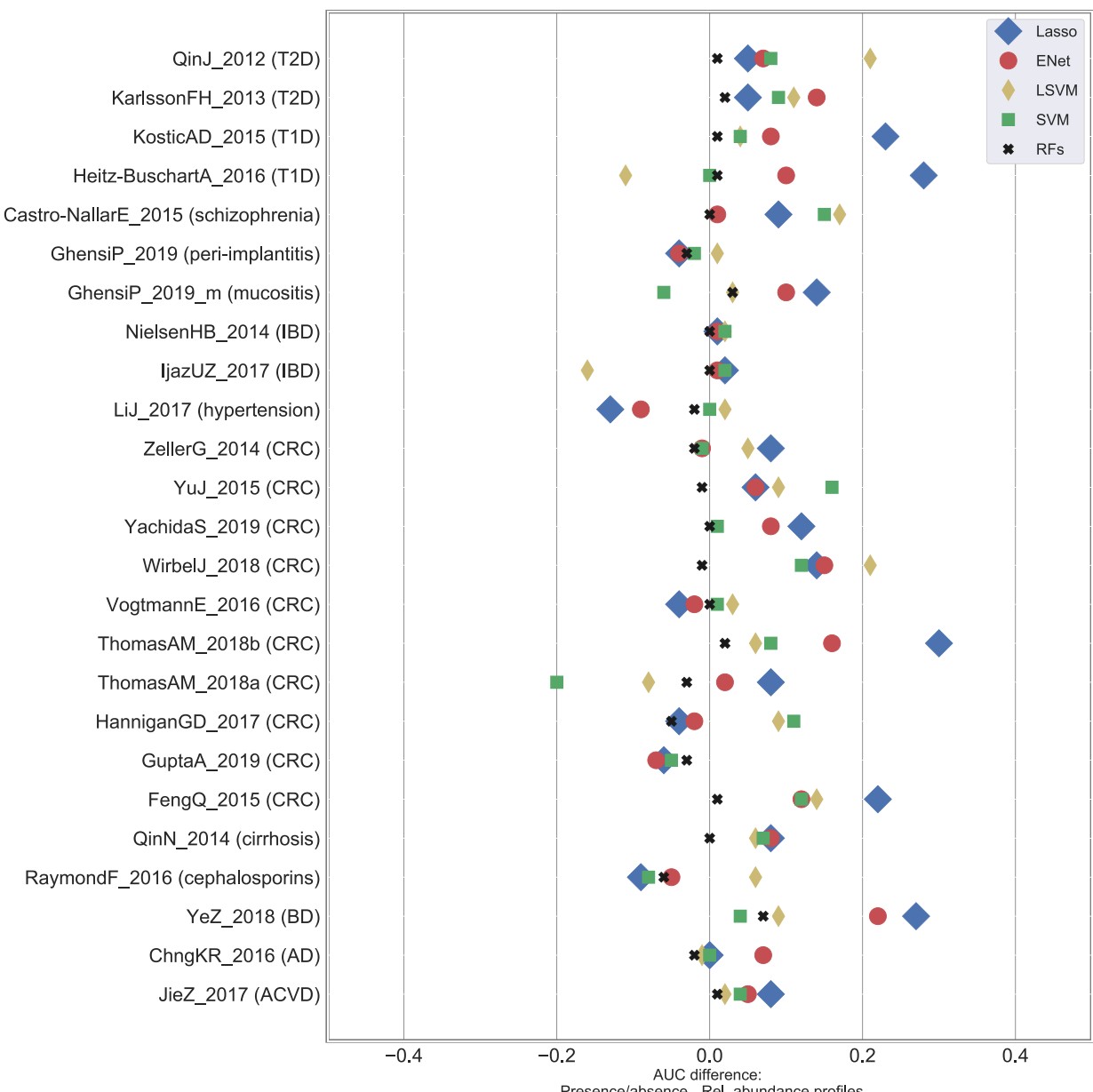

**Fig 5. Findings in terms of stability of the classification accuracy are robust to the classifier choice.** Differences in terms of AUC between presence/absence and relative abundance profiles for the 25 case-control shotgun datasets at varying classification algorithms. ENet: Elastic Net; LSVM: SVM with linear kernel; SVM: SVM with RBF kernel; RFs: Random Forests.

## Results and discussion

In this paper, we conducted a meta-analysis aiming at evaluating to which extent degradation from relative abundance to presence/absence of microbial taxa can impact host phenotype classification from human metagenomes. The analysis was conducted on 4,128 public available metagenomes coming from 25 datasets (**Table 1** and **Fig 1A**). Metagenomes were uniformly processed to generate species-level taxonomic profiles with MetaPhlAn3 [29] (see **Materials and Methods**) with metadata information available in the curatedMetagenomicData package

[30]. From relative abundance profiles, expressed as real numbers in the range [0, 1], we generated presence/absence profiles by simply thresholding the relative abundance values at 0%. This generated a set of boolean profiles where one indicated the presence of the species regardless of its relative abundance in the considered sample, while zero was associated with its absence.

## Baseline classification results replicate original findings

As baseline, we considered the classification approach that we originally proposed in [14] and that was then used for different tasks such as detection of microbial signatures linked to colorectal cancer (CRC) from human metagenomes [39], characterization of the oral microbiome in dental implant diseases [47], and identification of changes associated with dietary interventional studies [55]. More specifically, we considered a RF classifier applied on the species-level relative abundance profiles, and evaluated classification accuracies in terms of multiple metrics (i.e., area under the ROC curve (AUC), area under the precision-recall curve (AUPRC), precision, recall, and F1) using a cross-validation (CV) approach (see **Materials and Methods**). We obtained variable accuracies ranging from 0.56 (in terms of AUC) for hypertension in the LiJ_2017 dataset [44] to 0.99 for IBD in the IjazUZ_2017 dataset [45], with an average AUC across the 25 case-control studies equal to 0.83 (**S4 Table**). Such values were in line with what reported in the original publications, although a fair comparison is difficult to be performed due to differences in terms of adopted algorithms and input features. On the 17 publications that reported classification results on the same samples here considered, we obtained an average AUC of 0.80 in comparison to the average of 0.83 reported in the original publications (**S4 Table**).

## Degradation from species-level relative abundance to presence/absence profiles does not worsen classification accuracies

We applied the same classification approach on the same set of samples to the presence/absence profiles (**Materials and Methods**). In this way, we evaluated to which extent moving from relative abundance to presence/absence information could impact classification accuracies. Surprisingly, we observed negligible differences between the two experimental settings (**Figs 1B, 1C** and **S1** and **S2 Table**). In both cases (i.e., using presence/absence or relative abundance profiles), we obtained an average AUC of 0.83 (AUPRC = 0.83) across the 25 case-control studies, with AUC and AUPRC values strongly correlated (**S2 Fig**; Spearman correlation = 0.918). Some variations were observed at dataset-level (relative abundance outperformed presence/absence at a maximum of 0.06 in terms of AUC in the RaymondF_2016 dataset [34], while the opposite case was verified in YeZ_2018 [33] for an AUC difference of 0.07), however these were likely due to random perturbations and in none of the cases they were associated with statistically significant differences (p > 0.05, **S2 Table**). This was also confirmed in terms of the other metrics of comparison (i.e., precision, recall, and F1), with no significant differences between the two profile types (**S2 Table**). In a similar setting, we performed body site discrimination (oral vs stool samples) in the HMP dataset [10], with a value of AUC equal to 1.00 for both profile types. Therefore, such findings suggested that it was more the presence of same taxa rather than their actual relative abundance to be relevant for discrimination purposes.

We extended this analysis to 16S rRNA samples. More specifically, we considered the same set of 30 case-control studies for a total of 4,026 samples that were originally collected and analysed in [13] (**Fig 2A** and **S1 Table**). We applied the same pre-processing procedure adopted in [13] (**Materials and Methods**), and performed the prediction tasks by adopting the classification pipeline already considered for shotgun data. We obtained results similar to the ones presented in [13] on the genus-level relative abundance profiles (average AUC across the 30

datasets equal to 0.76 and 0.74 in our analysis and in [13], respectively) (**S5 Table**), although some differences could occur due to the different code implementations. By degrading relative abundance to presence/absence profile, we obtained few differences in the classification results between the two profile types (**Figs 2B, 2C and S4 and S5 Table**). Average AUC across the 30 studies was quite close (0.76 for relative abundance and 0.75 for presence/absence profiles), with differences that were statistically significant in only 3 out of 30 cases (**S5 Table**). Such differences, albeit impacting a limited number of datasets, may be due to the coarser taxonomic resolution and the higher noise component associated with 16S data.

## Statistically significant taxa are consistent between relative abundance and presence/absence profiles

We extended the analysis from classification to identification of differentially abundant/present taxa (i.e., possible biomarkers) through statistical testing (**Materials and Methods**). By comparing the sets of statistically significant species in the different case-control studies ($q < 0.05$; using Mann-Whitney U test for relative abundance and Fisher exact test for presence/absence profiles, both corrected through false detection rate (FDR), **S6 Table**) we found similar numbers (**Fig 1D and S7 Table**), with values more driven by disease and dataset types than average number of reads (**S5 Fig**). On average, we found 39 and 32 significant species from relative abundance and presence/absence profiles, respectively. We may hypothesize that diseases that rely on rare biomarkers are less affected by degradation to presence/absence profiles than the ones that are characterized by stronger community shifts in abundant and prevalent taxa. Although this is not sufficiently supported by our data, further investigation in this direction is warranted.

On a per dataset basis, p-values associated with statistically significant species correlated well between relative abundance and presence/absence profiles (**S6 Fig**). This was reflected also by the high percentage of taxa (78%) that were detected as significant in both cases, which was further confirmed by performing hierarchical clustering on the set of statistically significant taxa coming from relative abundance and presence/absence profiles (**S7 Fig**). Conversely, we identified discrepancies between case-enriched and control-enriched taxa in only 1.74% of the statistically significant features, which were coming from just 5 of the 24 analysed datasets (**S8 Fig**). Moreover, we didn't identify any taxa for which the two tests disagreed across datasets (**S8 Fig**).

Focusing on the gut microbiome datasets, we also identified the species that were mostly associated with disease or health (**S7 Fig**). The species most enriched in cases was *Clostridium bolteae* (significant in 78% of the diseases), followed by *Streptococcus anginosus group* (55%), *Ruthenibacterium lactatiformans* (55%), *Hungatella hathewayi* (55%), and *Eisenbergiella tayi* (55%) with all of them already reported in the literature as possible biomarkers for different disease conditions [6,13,39,41,56]. Similarly, species most enriched in controls were *Anaerostipes hadrus* (significant in 66% of the diseases), *Roseburia faecis* (55%), *Roseburia intestinalis* (55%), *Prevotella copri* (44%), and *Eubacterium hallii* (44%) [6,10,39,57].

Consistence between relative abundance and presence/absence outcomes was finally obtained on the 16S data, with 20 and 15 genera that were found to be significant on average from relative abundance and presence/absence profiles, respectively (**Fig 2D and S8 Table**).

## Relative abundance values lower than 0.001% do not impact classification outcomes

We evaluated how different values in thresholding relative abundance profiles could impact classification results. We thresholded the abundances at different values (i.e., moving from a

threshold equal to 0%—which corresponded to the presence/absence scenario discussed in the previous section—to 0.0001%, 0.001%, 0.01%, and 0.1%, **Materials and Methods**), meaning that values below the chosen threshold were forced to zero. We did not observe changes in the classification accuracy when the threshold was set to 0.0001% and 0.001% (**Figs 3A and S3A and S2 Table**). In both cases, we got an average AUC = 0.83 across the 25 case-control studies as obtained on the relative abundance profiles and using a threshold equal to zero, and no statistically significant differences were found. This was reflected by the number of statistically significant species (**Fig 3B and S7 Table**) that decreased very marginally from 32 (average value by considering 0% or 0.0001% as threshold) to 31 (threshold = 0.001%). Although very low abundant species may be actual biomarkers, they did not contribute to improving classification accuracies which was likely due to the impossibility to estimate their presence and relative abundance properly as being below or close to the limit of detection, which we quantified in this setting to be around 0.001% (with an average number of reads across our considered metagenomes equal to 47.5M). Major differences were obtained when thresholding at higher values (i.e., 0.01% and 0.1%). In these cases, average AUC decreased to 0.81 (threshold = 0.01%) and 0.78 (threshold = 0.1%), with significant differences in 3 and 6 cases, respectively.

Results on rarefied reads (**Materials and Methods**) showed, as expected, a slight decrease in terms of classification accuracies and number of detected biomarkers with respect to the original data set, although patterns in function of the thresholding value when going from relative abundance to presence/absence data were confirmed (**S9 Table**).

## Coarser taxonomic levels are less robust to profile degradation

We further tested to which extent classification accuracy was affected by the taxonomic resolution level considered to feed the classifier. By considering original relative abundance profiles, average AUC moved from 0.83 (species-level resolution) to 0.80 (with 3 statistically significant cases), 0.78 (6), and 0.76 (11) for genus, family, and order levels, respectively (**S10 Table**). Such differences, albeit not too strong, suggested species as "optimal" level to optimize classification accuracies, with further improvements that may be obtained—although not tested here due to methodological limitations—with sub-species- or strain-level resolutions.

Similarly, we compared classification accuracies between relative abundance and presence/absence profiles at different taxonomic levels. While no differences were obtained at species-level (as already discussed in **Fig 1**), we observed that coarser resolutions brought increasing AUC differences (**Figs 4 and S3B and S11 Table**). An average AUC difference of 0.022, 0.041, and 0.061 was obtained for genus, family, and order, respectively (with 0, 1, and 2 statistically significant cases, respectively). Similar patterns were observed in terms of number of statistically significant features (**S7 Table**).

## Findings are robust to cross-study analysis and to the classifier choice

We applied the same approach on a cross-study setting. We considered the ten independent metagenomics studies associated with CRC for a total of 1313 samples (**Table 1**) and applied a leave-one-dataset-out (LODO) approach in which the model was built on all datasets but the single dataset used for testing (**Materials and Methods**). As previously reported [39,41], we observed an overall moderate decrease of the accuracy when moving from CV (average AUC equal to 0.80) to LODO (average AUC equal to 0.76; **S9 Fig and S12 Table**). More importantly, we confirmed previous findings in terms of stability of the accuracy when moving from relative abundance to presence/absence profiles at species-level resolution (**Fig 6A**). The average AUC remained stable at 0.76 for the presence/absence profiles at threshold equal to 0%, 0.0001%, and 0.001%, while it decreased to 0.74 and 0.73 when thresholding at 0.01% and

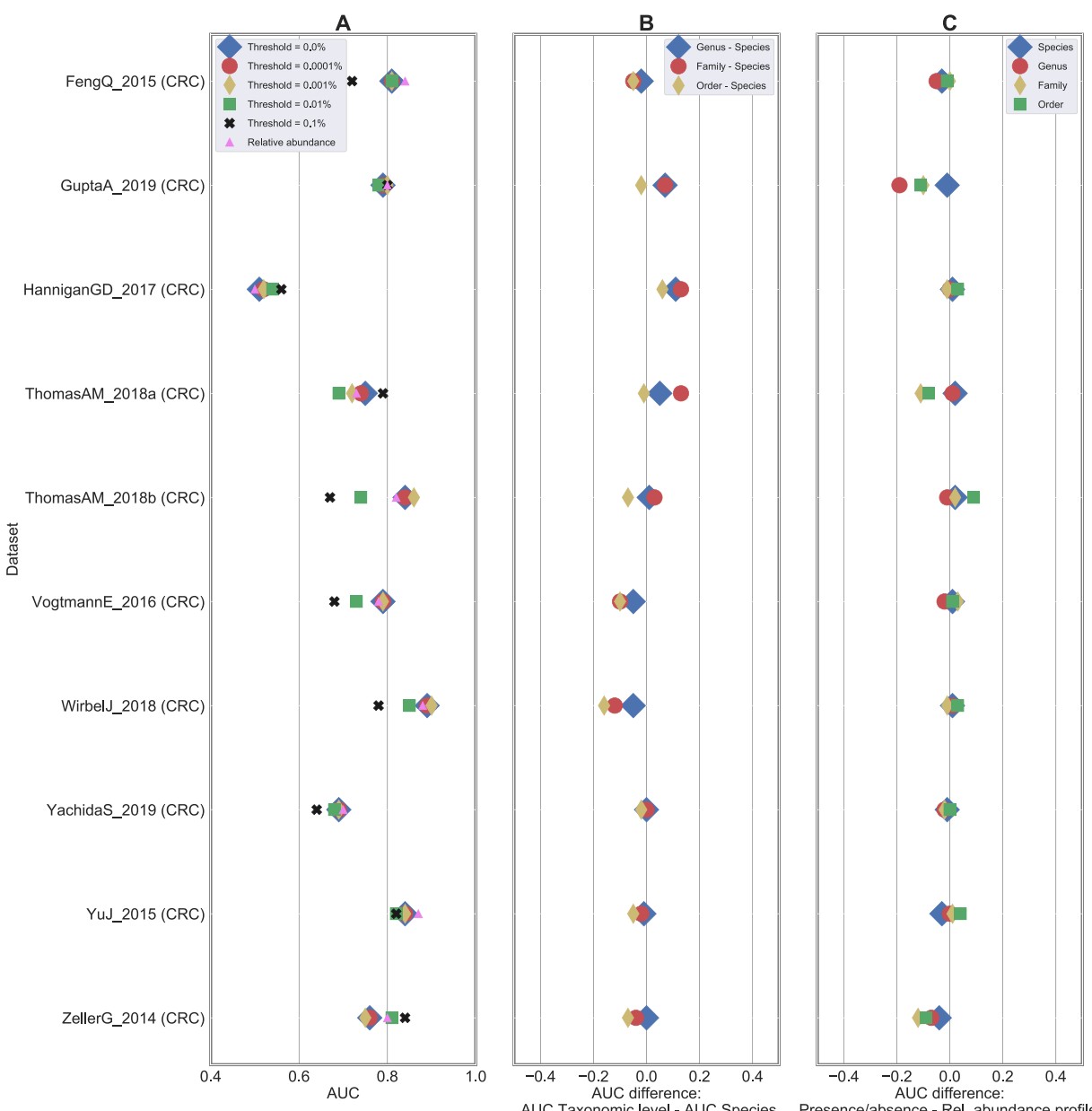

**Fig 6. Degradation of relative abundance profiles does not impact LODO classification.** Results in terms of leave-one-dataset-out (LODO) validation on 10 CRC shotgun datasets. (**A**) AUC scores using RF as back-end classifiers on species-level relative abundance (in pink) and presence/ absence profiles generated at different threshold values. (**B**) Difference in AUC between species and other taxonomic-level resolutions. A negative value indicates that species-level outperforms the comparison level. (**C**) Difference in AUC between presence/absence and relative abundance classification results at varying taxonomic levels.

0.1%, respectively. We also confirmed that the better taxonomic resolution was associated with smaller classification performance differences between relative abundance and presence/ absence data (**Fig 6B and 6C**).

We finally tested if the choice of the classification method could impact the findings described in the previous sections in terms of degradation from relative abundance to presence/absence profiles. First, we confirmed [14] the superiority of RF with respect to other four classification methods (i.e., Lasso [58], Elastic Net [59], and support vector machines (SVMs)

with linear and RBF kernels [60]) on both relative abundance (**S10A Fig** and **S13 Table**) and presence/absence profiles (**S10B Fig** and **S13 Table**), and this was also verified on 16S rRNA data (**S3 Table**). On average, thresholding of relative abundance values did not negatively impact classification accuracies, instead it generally improved results in a quite unexpected way (**Figs 5** and **S3C**). Higher differences were observed for Lasso, with an average AUC equal to 0.79 and 0.72 for presence/absence and relative abundance data, respectively, and the same pattern was obtained for the other classifier methods (with an average difference in terms of AUC equal to 0.05, 0.03, and 0.02 for ENet, LSVM, and SVM, respectively). We observed a greater variability of the classification accuracies with respect to what was observed for RF classification. In fact, we obtained statistically significant differences in Lasso, ENet, LSVM, and SVM studies for 10, 6, 5, and 6, respectively, however always in majority in favour of the presence/absence data. We therefore conclude that, despite a few differences occurred in a limited number of cases, maximization of classification accuracies was generally made possible through presence/absence profiles.

## Conclusions

In the present study, we conducted a meta-analysis on 25 publicly available datasets spanning more than 4,000 shotgun metagenomes and associated with different case-control studies. By applying species-level taxonomic profiling and machine-learning based classification approaches based on state-of-the-art methodologies we demonstrated that the presence of microbial taxa is sufficient to maximize classification accuracies. This was accomplished by degrading original relative abundance data to presence/absence profiles by considering different threshold values. We estimated a value of 0.001% in terms of relative abundance as limit of detection, meaning that although very low abundant species may be actual biomarkers they were not useful to improve classification accuracy. Results were robust to the choice of the classifier. This was obtained by considering different traditional classification algorithms that are designed for continuous data and potentially "suboptimal" when applied on binary data. This actually reinforces our findings, meaning that accuracies may be even better when models on presence/absence profiles are trained using classifiers more designed for binary data. Moreover, although doing an extensive evaluation of existing classifiers is out of the scope of the present study, maximization of classification accuracies may be reached by adopting other classification approaches including the ones specifically proposed for microbiome data analysis [61,62]. Findings were finally extended from cross validation to cross study analysis and confirmed on 16S rRNA data associated with a compendium of more than 4,000 samples coming from 30 public studies.

The growing literature aiming at identifying microbial biomarkers for different diseases opened the possibility to build non-invasive diagnostic tools from microbiome data. To this purpose, much superior accuracy can be achieved by considering multi-feature rather than single biomarkers diagnostic models, and in which machine learning-based classification approaches have a fundamental role in building such models. Moreover, maximal accuracy can usually be achieved by using a limited number of features (in the order of ten or twenty). Such findings recently presented in the literature in addition to outcomes of our study, which suggest that the detection of microbial taxa is sufficient to maximize classification accuracies, are important steps toward the development of fast and inexpensive tests applied on stool samples for diagnostic purposes.

## Supporting information

**S1 Table. Summary of the 30 classification tasks derived from 16S rRNA datasets for case-control prediction.** ASD: Autism spectrum disorder, CD: Crohn disease, CDI: Clostridium

difficile infection, CIRR: Cirrhosis, MHE: Minimal hepatic encephalopathy, CRC: Colorectal cancer, EDD: enteric diarrheal disease, HIV: human immunodeficiency virus, NASH: non-alcoholic steatohepatitis, OB: obesity, PAR: Parkinson's disease, PSA: psoriatic arthritis, RA: Rheumatoid arthritis, T1D: type-1 diabetes, UC: ulcerative colitis. Non-CDI controls are patients with diarrhea who tested negative for *C. difficile* infection.
(XLSX)

**S2 Table. Results obtained from the classification process done on the shotgun datasets.** Comparison in terms of AUC, AUPRC, F1, precision, recall between relative abundance and presence/absence profiles at different threshold levels. Results are obtained using RF classification at the species-level taxonomic resolution.
(XLSX)

**S3 Table. Comparison in terms of AUC between relative abundance and presence/absence profiles with different classification algorithms (RF: Random Forest; Lasso; ENet: Elastic Net; LSVM: SVM with linear kernel; SVM: SVM with RBF kernel).**
(XLSX)

**S4 Table. Comparison in terms of AUC between our results (using RF classification on the relative abundance profiles) and the ones reported in the original publications.** In most of the cases, different classifier algorithms and/or input features were used in the original analysis. Original papers that did not conduct a classification analysis are not included in this table.
(XLSX)

**S5 Table. Results obtained from the classification process done on the 16s datasets.** Comparison in terms of AUC, AUPRC, F1, precision, recall between relative abundance and presence/absence profiles at different threshold levels. Results are obtained using RF classification at the species-level taxonomic resolution.
(XLSX)

**S6 Table. P-values (after FDR correction) obtained by testing differences in abundance of each species between controls and cases.**
(XLSX)

**S7 Table. Number of statistically significant taxa (q< = 0.05) between cases and controls for each shotgun dataset and at varying input features (relative abundance vs presence/absence profiles) and taxonomic level.**
(XLSX)

**S8 Table. Number of statistically significant taxa (q< = 0.05) between cases and controls for each 16s dataset and at varying input features (relative abundance vs presence/absence profiles).**
(XLSX)

**S9 Table. Results obtained on three selected shotgun datasets after rarefying metagenomes at 1M reads.** Comparison in terms of AUC, F1, precision, recall, in addition to number of statistically significant taxa (q< = 0.05), between the results obtained classifying on the abundances matrix and the classification made on the presence/absence boolean matrix at different taxonomic levels (only at species level).
(XLSX)

**S10 Table. Results obtained from the classification process done on the shotgun datasets.** Comparison in terms of AUC between the results obtained classifying at different taxonomic

resolution levels. The results are obtained using the RF classifier on the relative abundances matrixes.
(XLSX)

**S11 Table. Results obtained from the classification process done on the shotgun dataset.** Comparison in terms of AUC, F1, precision, recall between the results obtained classifying on the abundances matrix and the classification made on the presence/absence boolean matrix at different taxonomic levels (species, genus, etc).
(XLSX)

**S12 Table. Results obtained by the LODO classification for datasets associated with CRC.** Comparison in terms of AUC obtained classifying thresholding the dataset at different levels and at different taxonomic levels.
(XLSX)

**S13 Table. Comparison in terms of AUC, F1, precision, recall between the results obtained from different classifiers on the relative abundances matrix and on the presence absence boolean matrix (only at species level).**
(XLSX)

**S1 Fig. Classification accuracies are robust to degradation from species-level relative abundance to presence/absence profiles in shotgun datasets.** Comparison in terms of AUC between presence/absence and relative abundance profiles for the 25 case-control shotgun datasets.
(PNG)

**S2 Fig. AUC correlates well with AUPRC.** Comparison in terms of classification accuracies between AUC (area under the curve) and AUPRC (area under the precision-recall curve) for the 25 case-control shotgun datasets and by considering relative abundance (in blue; Spearman correlation = 0.889) and presence/absence (in red; Spearman correlation = 0.918) profiles.
(PNG)

**S3 Fig. Classification accuracies are robust to degradation from species-level relative abundance to presence/absence profiles in shotgun datasets.** Comparison in terms of AUC between presence/absence and relative abundance profiles for the 25 case-control shotgun datasets by (**A**) thresholding at different relative abundance values (ranging from 0% to 0.1%), (**B**) changing taxonomic resolution (from species to order level), and (**C**) changing classification algorithm.
(PNG)

**S4 Fig. Classification accuracies are robust to degradation from species-level relative abundance to presence/absence profiles in 16S rRNA datasets.** Comparison in terms of AUC between presence/absence and relative abundance profiles for the 30 case-control 16 rRNA datasets.
(PNG)

**S5 Fig. Number of differentially abundant species has weak correlation with the average number of reads.** Each dot represents one of the 26 case-control shotgun studies. The number of statistically significant species is computed on relative abundance profiles.
(PNG)

**S6 Fig. P-values associated with statistically significant species correlate well between relative abundance and presence/absence profiles.** Each dot represents a different taxa (i.e.,

species) and we report only species significant in at least one of the two data types. Only datasets with at least ten data points are shown.
(PNG)

**S7 Fig. Statistically significant taxa are consistent between relative abundance and presence/absence data on a per dataset basis.** Heatmap generated on the p-values (after FDR correction; p > 0.05 in grey) obtained by applying statistical tests on the case-control metagenomic datasets. Only the 18 datasets with at least one discriminative taxa are reported. Left-most colorbar identifies the taxonomic class of each taxa. The two right-most colorbars indicate the percentage of diseases for which the species resulted to be enriched in controls (in green) and in cases (in red). This percentage is computed on a per disease basis, when multiple datasets are available for the same disease, the taxa is considered significant when detected as significant in at least one dataset.
(PNG)

**S8 Fig. Statistically significant taxa from relative abundance and presence/absence profiles did not disagree across datasets.** We identified discrepancies between case-enriched and control-enriched taxa derived from relative abundance and presence/absence data in only 1.74% of the statistically significant features, which were coming from just 5 datasets. No taxa disagreed across datasets.
(PNG)

**S9 Fig. Degradation of relative abundance profiles has a limited impact on both CV and LODO classification.** AUC scores using RF as back-end classifiers on species-level relative abundance and corresponding presence/absence profiles in CV and LODO settings.
(PNG)

**S10 Fig. RFs generally outperform other classifiers.** Results on the 25 case-control shotgun studies by considering different classification algorithms. Difference in AUC between RFs and other classification methods on (**A**) the relative abundance and (**B**) the presence/absence profiles. A positive value indicates that the comparison method outperforms RFs.
(PNG)

## Author Contributions

**Conceptualization:** Edoardo Pasolli.

**Data curation:** Renato Giliberti.

**Formal analysis:** Renato Giliberti, Edoardo Pasolli.

**Funding acquisition:** Danilo Ercolini, Edoardo Pasolli.

**Investigation:** Renato Giliberti.

**Methodology:** Renato Giliberti, Edoardo Pasolli.

**Project administration:** Edoardo Pasolli.

**Software:** Renato Giliberti, Edoardo Pasolli.

**Supervision:** Edoardo Pasolli.

**Validation:** Renato Giliberti, Sara Cavaliere, Italia Elisa Mauriello.

**Visualization:** Renato Giliberti.

**Writing – original draft:** Renato Giliberti, Danilo Ercolini, Edoardo Pasolli.

**Writing – review & editing:** Renato Giliberti, Danilo Ercolini, Edoardo Pasolli.

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
