## [Decision Letter · Decision Letter 0]

8 Dec 2021

Dear Dr Pasolli,

Thank you very much for submitting your manuscript "Host phenotype classification from human metagenomes is more driven from presence than abundance of microbial taxa" for consideration at PLOS Computational Biology.

As with all papers reviewed by the journal, your manuscript was reviewed by members of the editorial board and by several independent reviewers. In light of the reviews (below this email), we would like to invite the resubmission of a significantly-revised version that takes into account the reviewers' comments.

As you can see from the reviews, the feedback is generally positive and, in our judgement, none of the raised issues present a fundamental objection to the soundness and relevance of the work. We nonetheless ask that the authors consider all the points raised to improve the manuscript.

We cannot make any decision about publication until we have seen the revised manuscript and your response to the reviewers' comments. Your revised manuscript is also likely to be sent to reviewers for further evaluation.

Sincerely,

Luis Pedro Coelho

Associate Editor

PLOS Computational Biology

Ilya Ioshikhes

Deputy Editor

PLOS Computational Biology

As you can see from the reviews, the feedback is generally positive and, in our judgement, none of the raised issues present a fundamental objection to the soundness and relevance of the work. We nonetheless ask that the authors consider all the points raised to improve the manuscript.

Reviewer's Responses to Questions

**Comments to the Authors:**

Reviewer #1: General:

Giliberti and colleagues conduct a machine learning meta-analysis including different metagenomic datasets, using the information about presence and absence of bacterial species as input features for modelling rather than their relative abundance. The machine learning analyses are performed in a technically thorough manner and uncover a surprising finding, i.e. that the presence/absence information alone seems to be as predictive for various disease states as relative abundances. However, the biological relevance of the presented results are not fully clear and should be discussed in more detail.

Major:

1. The authors discuss the implications for practical applications in diagnostic tests. However, it remains unclear to this referee whether detection of bacteria present rather than their quantification (by a cheap target approach such as qPCR)would make a diagnostic routine easier to implement in practice. The authors might want to discuss for which diagnostic assays this could matter.

2. We feel that some of the chosen visualizations are not optimal to convey differences between various approaches in terms of AUC differences. A better alternative might be scatter plots with the original AUC on the x-axis and the AUC from the model trained on presence/absence data (or with different threshold, different taxonomic levels, or different machine learning algorithms) on the y-axis.

3. The included datasets (and samples within a dataset) vary considerably in terms of their mean sequencing depth, which likely affects the detection of taxa with lower relative abundance. For this reason data is often downsampled (rarefied) before presence/absence calls and derived richness estimates are calculated. This issue is partially addressed by the authors’ use of different threshold values above which the authors call a taxa “present”, but this aspect might need further clarification. A more explicit description of how their thresholding strategy relates to downsampling datasets should be added and in the best case show both approaches could be compared empirically (on a few datasets).

4. The analysis of statistically significant taxa in both approaches (lines 330 and following, also SFig. 4) feels disconnected from the rest of the manuscript and generally underdeveloped. Could the authors compare the p-values from the Wilcoxon and Fisher tests, maybe again via scatter plots (on -log10 scale of the p-values)? How well do the p-values correlate? Are there taxa for which the two tests disagree consistently across datasets? How does it look like when abundance fold change and prevalence difference across groups is compared? Is there a clear correlation? In general, this analysis could be greatly expanded to bolster the biological interpretation of the results.

In this context, it is also important to note that there are many CRC datasets in the set of included studies, potentially biasing the disease-associations reported in lines 339 and following. Instead, the authors could try to get a single significance value for each disease (including all available studies of this disease) and compare again.

5. Does the disease which is to be predicted have any influence on loss of/retaining accuracy after converting features to presence/absence information? One could imagine that CRC, relying more on rare biomarkers, could be less affected by downgrading features to presence/absence information than for example IBD, which is characterized by stronger community shifts in highly abundant and prevalent taxa. This analysis could contribute to developing a clearer understanding of the biological relevance of the presented analysis.

Minor:

1. The LODO analysis for CRC studies could be moved from the supplement to the main manuscript, especially if the display items are streamlined via scatter plots.

2. Are the different models trained on the same cross-validation splits? If not, adopting this approach could remove the random noise introduced by repeated CV splitting.

Reviewer #2: Giliberti et al. present a very well written paper comparing the predictive performance of quantitative vs qualitative taxonomic profiles. The breadth of the work is comprehensive in the range of phenotypes (IBD, CRC, T2D, etc) being analyzed and the data types (16S rRNA and Metagenomics). The core question being answered is an interesting one and this paper will likely be of interest to the broad microbiome community. However, there are three main issues that would first need to be addressed. The first is a logistical issue concerned with the availability of the source code used to generate the data in this paper. The second issue is concerned with the use of the AUC of the ROC to evaluate classification tasks on unbalanced datasets. The last issue is concerned with the choice of test used to compute differential abundance.

The code used to generate the data and figures included in this paper needs to be made available and well documented. Please note the versions of the software dependencies used, how to run these scripts, and when and where the datasets were downloaded from. My recommendation is to host this repository on GitHub or another source code sharing platform.

Using the AUC of the ROC for evaluating the performance of a classification task for unbalanced datasets is likely to yield misleading results. My suggestion would be to use this metric and the area under the precision-recall curve AUPRC. Alternatively using any of the other metrics that were already measured for this dataset would also be appropriate. These metrics need to be also evaluated with the statistical testing framework used for the AUC of the ROC.

Lastly, performing differential abundance comparisons using Mann-Whitney’s U test is inappropriate due to the compositional nature of the data. The authors should update these results to use a different test and or normalization strategy

Beyond these issues, I would recommend that the authors rethink the title of their paper, as it stands the classification performance is shown to be comparable in between both cases. The title makes an implication about qualitative profiles outperforming quantitative profiles. Furthermore the use of the term "metagenome" in the title is misleading since amplicon sequencing data is also presented in this paper.

Reviewer #3: This paper proposed a meta-analysis on 25 publicly available dataset of metagenomic studies and 30 public dataset of 16S rRNA studies, the major investigation purpose is whether presence/absence data could be a valid and efficient indicator for classifying samples. The conclusion is that by degrading abundance data to binary(presence/absence) data, the classification performance could be maintained without decreasing AUC in many parameter settings. This is an interesting paper for host-phenotype classification by only considering binary(presence/absence ) data instead of abundance data, because it may suggest that it may be possible for designing microbe-based diagnostic tasks in future by the detection of the presence of a microbial taxa set rather than the complex abundance estimation by sequencing technology. The writing of the paper is very clear, however there are several concerns may be useful to consider, at least valuable to discuss:

(1)The paper used Random forest, SVM, Lasso and Elastic Net for comparison, it is possible to adopt some updated classification approaches, for example:

https://pubmed.ncbi.nlm.nih.gov/32396115/

https://pubmed.ncbi.nlm.nih.gov/32657370/

(2)The paper designed many experiments by using a lot of datasets, it is useful to provide a website link for downloading the processed dataset for comparison and reproduce the results.

(3)Traditionally, classification methods including Random forest, SVM, Lasso and Elastic Net are designed for continuous data instead of binary data (presence/absence data). When the data form is changed, the algorithm may not adapted to it. The author may explain why these methods could work on both continuous data (abundance data) and binary data (presence/absence data)

**Have the authors made all data and (if applicable) computational code underlying the findings in their manuscript fully available?**

Reviewer #1: Yes

Reviewer #2: **No: **

Reviewer #3: Yes

PLOS authors have the option to publish the peer review history of their article (what does this mean?). If published, this will include your full peer review and any attached files.

Reviewer #1: No

Reviewer #2: No

Reviewer #3: **Yes: **Xingpeng Jiang
---

## [Editor Report · Decision Letter 1]

20 Mar 2022

Dear Dr Pasolli,

Thank you very much for submitting your manuscript "Host phenotype classification from human microbiome data is mainly driven by the presence of microbial taxa" for consideration at PLOS Computational Biology. We are likely to accept this manuscript for publication, providing that you modify the manuscript according to the recommendations below:

We consider that the scientific questions have been addressed in this revision. However, before accepting the manuscript, we ask that the authors revise the figures, particularly the supplemental ones, for improved readability and accessibility:

- The fonts and markers are often too small to read comfortably even though there appears to be space available. This is particularly true in the supplemental figures, but some of the main figures could also be improved.

- We also ask that Figs S7 and S8 be redone using a coulorblind-safe color scheme rather than rely on red/green distinction.

- We ask the authors to consider pointing to Figs. S1 and S2 from the caption of Fig 1.

Sincerely,

Luis Pedro Coelho

Associate Editor

PLOS Computational Biology

Ilya Ioshikhes

Deputy Editor

PLOS Computational Biology

[LINK]

We consider that the scientific questions have been addressed in this revision. However, before accepting the manuscript, we ask that the authors revise the figures, particularly the supplemental ones, for improved readability and accessibility:

- The fonts and markers are often too small to read comfortably even though there appears to be space available. This is particularly true in the supplemental figures, but some of the main figures could also be improved.

- We also ask that Figs S7 and S8 be redone using a coulorblind-safe color scheme rather than rely on red/green distinction.

- We ask the authors to consider pointing to Figs. S1 and S2 from the caption of Fig 1.

Figure Files:

Data Requirements:

Reproducibility:

References:

---

## [Editor Report · Decision Letter 2]

29 Mar 2022

Dear Dr Pasolli,

We are pleased to inform you that your manuscript 'Host phenotype classification from human microbiome data is mainly driven by the presence of microbial taxa' has been provisionally accepted for publication in PLOS Computational Biology.

Best regards,

Luis Pedro Coelho

Associate Editor

PLOS Computational Biology

Ilya Ioshikhes

Deputy Editor

PLOS Computational Biology

---

## [Editor Report · Acceptance letter]

18 Apr 2022

PCOMPBIOL-D-21-01860R2 

Host phenotype classification from human microbiome data is mainly driven by the presence of microbial taxa

Dear Dr Pasolli,

I am pleased to inform you that your manuscript has been formally accepted for publication in PLOS Computational Biology. Your manuscript is now with our production department and you will be notified of the publication date in due course.

With kind regards,

Livia Horvath
